# The New Frontier in Oxytocin Physiology: The Oxytonic Contraction

**DOI:** 10.3390/ijms21145144

**Published:** 2020-07-21

**Authors:** Claudia Camerino

**Affiliations:** 1Department of Biomedical Sciences and Human Oncology School of Medicine, University of Bari Aldo Moro, 70100 Bari, Italy; ccamerino@libero.it; 2Department of Physiology and Pharmacology “V. Erspamer”, SAPIENZA University of Rome, 00185 Rome, Italy

**Keywords:** oxytocin, cold stress, skeletal muscle, obesity, heart, Prader–Willy syndrome

## Abstract

Oxytocin (Oxt) is a nine amino acid peptide important in energy regulation and is essential to stress-related disorders. Specifically, low Oxt levels are associated with obesity in human subjects and diet-induced or genetically modified animal models. The striking evidence that Oxt is linked to energy regulation is that Oxt- and oxytocin receptor (Oxtr)-deficient mice show a phenotype characterized by late onset obesity. Oxt−/− or Oxtr−/− develop weight gain without increasing food intake, suggesting that a lack of Oxt reduce metabolic rate. Oxt is differentially expressed in skeletal muscle exerting a protective effect toward the slow-twitch muscle after cold stress challenge in mice. We hypothesized that Oxt potentiates the slow-twitch muscle as it does with the uterus, triggering “the oxytonic contractions”. Physiologically, this is important to augment muscle strength in fight/flight response and is consistent with the augmented energetic need at time of labor and for the protection of the offspring when Oxt secretion spikes. The normophagic obesity of Oxt−/− or Oxtr−/− mice could have been caused by decreased skeletal muscle tonicity which drove the metabolic phenotype. In this review, we summarized our findings together with the recent literature on this fascinating subjects in a “new *oxytonic* perspective” over the physicology of Oxt.

## 1. Introduction

According to the principle of “maximum parsimony” enunciated by Galileo Galilei as “Nature does not work with many things what it can operate with few,” (dialogue concerning the two chief world systems, Galileo Galilei, 1632) means that every natural phenomenon is always realized with the minimum expenditure of both matter and energy [1,2,3]. Its analytical formulation consists in the principle of minimum work. Stretching this concept into physiology, let us hypothesize that the same matter, e.g., a hormone/neurotransmitter can have de facto two or more functional identities according to its site of synthesis or target organ. The recent progress in genetically manipulated animal models widened this discipline advancing our knowledge of “whole organism physiology,” versus a more cell-specific concept [4]. From an evolutionary point of view, the biological advantage of this pleiotropic mechanism is that the lack of effects of a specific gene following challenging situations as pathophysiological conditions or gene downregulation after cold stress challenge for example [5] can be counterbalanced by the action of other genes [5,6]. This brief explanation is to introduce that in this review, we will present data that Oxt has several effects different from the well-known effects on uterocontractility and lactation [7,8]. These effects include the regulation of energy and metabolism, appetite regulation, and effects on the gastrointestinal system, skeletal, and cardiac muscle [9]. In this review, a special emphasis will be given to the effects of Oxt on skeletal muscle and to our recent paradigm shifting hypothesis that Oxt may increase the tone of the slow-twitch muscle upregulating its receptor and triggering the “oxytonic contraction” after cold stress challenge in mice [10,11].

## 2. Oxytocin- and Oxytocin Receptor-Deficient Mice: The Paradox of Normophagic Obesity

About 10 years ago or so, in our laboratory raised the notion that the hormone/neurotransmitter Oxt is related to the regulation of energy and metabolism. Oxt is synthesized as, prohormone, that is processed to generate the biologically active amidated nonapeptide [12]. The magnocellular axons of the paraventricular nuclei (PVN) and supraoptical nuclei (SON) in the hypothalamus secrete oxytocin (Oxt), in the posterior pituitary, and release Oxt, to the bloodstream [13,14]. Oxt acts as a hormone regulating functions of peripheral target organs and as a neurotransmitter [15]. Genetic studies in humans, gene knockout experiments in mice, and pharmacological manipulation showed that Oxt regulates a diversity of social behaviors related to reproduction. Indeed, Oxt concentration increases during challenging situations including pregnancy and lactation in females. Oxt triggers aggressive behavior that is important after labor for the protection of the offspring when the offspring is most vulnerable to predators, and Oxt concentration in plasma is at its peak [15,16,17,18]. Consistent with these data, we hypothesized that Oxt may increase muscle tone to ensure a better response to the “fight response” and triggering “the oxytonic contraction.” On this regard we showed that Oxtr expression increases in soleus muscle after cold stress challenge in mice [10]. The Oxt system is implicated also in the etiology of Prader–Willy syndrome, autistic spectrum disorder, and Schaaf-Yang syndrome [19]. Moreover Prader–Willy syndrome patients are hypotonic at birth and present low Oxt concentration in plasma consistent with our hypothesis [19].

Oxt is structurally similar to vasopressin and is synthesized also in site of synthesis outside the nervous system like the gastrointestinal tract and bone [20,21]. Oxt secretion is pulsatile, and estrogens augment Oxt effects by increasing the expression of Oxtr [22]. In this review, we will explore the several effects of Oxt besides pregnancy. Indeed, Oxt is known to contract and potentiate the uterus more than 200 times during labor and to contract the smooth muscle of breast favoring milk ejection [7,8]. Oxt is also involved in energy regulation and is an anorexigenic hormone that promote weight loss by decreasing energy intake. Consistent with this data in mice, in human studies, Oxt concentrations are positively associated with lean mass [9].

Interestingly mice that are homozygous for deletions of either Oxt or its receptor develop late onset obesity and metabolic syndrome. The glimpse of something missing in the Oxt- and Oxtr-deficient mice was the fact that their metabolic phenotype matured in the absence of hyperphagia. This is in contrast to the expectation that hypothalamic Oxt decreases food intake by increasing leptin concentration in plasma [23,24]. Moreover, the metabolic role of Oxt diverges in young versus older animals or alternatively it takes time to reach full force. This concept was named in our laboratory “the oxytocin paradox.” Several explanations have been given to this discrepancy including that Oxt may only mark the identity of neurons projecting from PVN but its action is mediated by classical neurotransmitters like GABA, alternatively Oxt may be anorexigenic in normal mice, but developmental mechanisms may compensate for its absence in Oxt−/− or Oxtr−/− mice [11,25,26]. Another hypothesis was that the appetite of Oxt−/− reported as normal, in spite of the hyperleptinemia, was excessive relative to the degree of adiposity [27]. This hypothesis was ruled out by the evidence that the stomachs of Oxt-deficient mice were reported comparable to wild-type mice for size and weight, which exclude any excess in food consumption [24]. Oxtr-deficient mice are thermogenically impaired, with a basal temperature lower than wild-type, which shed a light on the role of Oxt on temperature regulation and lean/fat mass composition in this model [23]. However, the lean/fat mass composition in skeletal muscle could be the reason for the normophagic obesity in this model. Indeed it took us about 10 additional years of study to come to the conclusion that the effects of Oxt on metabolism and energy are both direct, as Oxt is anorexigenic, and indirect, as Oxt acts specifically on muscles potentiating the majority of the slow-twitch muscles as it does with the uterus [11,23]. The normophagic obesity of Oxt KO mice was probably caused by a general muscular loss of function that slowly increased the intramuscular adipose tissue and ectopic fat accumulation in skeletal muscle and ultimately drove the late onset obesity and metabolic phenotype rather than by increased food consumption. The presence of concomitant sarcopenia and obesity confers worse functional outcome compared to either alone. Nevertheless, the study of Oxt in skeletal muscle and fat accumulation needs a further investigation. Studies on genetic models of obesity highlight that nutritional status does not always determine Oxt concentrations in blood. For example, in ob/ob mice, which are homozygous for leptin expression, no difference in serum Oxt was detected relative to wild type, whereas in db/db mice, which are leptin-resistant because they lack the long isoform of the leptin receptor Ob-Rb, serum Oxt concentrations were decreased relative to lean control mice [28,29]. Pharmacological administration of Oxt caused a loss in body weight higher than expected based on the decrease in food consumption, suggesting that Oxt may also increase energy expenditure, although Oxt administration does not affect physical activity directly. Of note, it was also interesting that exogenous administration of Oxt improves sarcopenia and muscle mass [30].

## 3. Thermoregulation after OXT-Neuron Depletion

The PVN is a critical locus of energy balance control. Oxt-expressing neurons in PVN are involved in regulating energy balance [31,32,33]. In a recent work, diphtheria toxin was injected into mice having the Oxt promoter driving cre-expression and the cre-inducible diphtheria toxin receptor to ablate Oxt-expressing neurons in PVN and investigate the role of Oxt in energy balance. The Cre-lox recombination approach is commonly used to generate cell-specific gene inactivation or activation [34]. This injection causes a significant decrease of Oxt neurons in PVN. Interestingly Oxt ablation did not alter food intake nor body weight at room temperature at either chow or high-fat diet. Differently, when exposed to thermogenic stress at 4 °C for 3 h, the murine model in which Oxt neurons have been pharmacologically ablated had lower core body temperature, a decreased response of brown adipose tissue to cold but increased heat at the skin [35] meaning that temperature at the core diverges from temperature at the surface in these mice. So, to adapt the body to low temperatures, Oxt neurons increase BAT activity and decrease peripheral vasodilation to reduce energy loss. In sum, Oxt-neurons ablation shows no role for Oxt-neurons in energy regulation at neutral temperature but these findings show the role of Oxt-neurons and Oxt in regulating the thermogenic response to cold [35]. Nevertheless, the role of Oxt-expressing neurons in energy regulation is not clear yet [36,37]. Although initial work on Oxt KO reported no metabolic phenotype, recent work showed late onset obesity in Oxtr and Oxt KO mice [23,24]. This obesity was initially explained as the result of altered energy expenditure rather than altered food intake [23,38]. Thus, in addition to decreasing food consumption, Oxt may be involved in regulating the lean/fat composition of the body. Oxt may increase muscle tone. This is finalized to a more powerful physical activity in time of need during the perinatal period, for the protection of the offspring, or any challenging situation that requires competition within pairs like for reproduction or food, whereas the lack of Oxt may cause the infiltration of skeletal muscle with fat. The ablation of Oxt-neurons by diphtheria toxin injection causes also an alteration of metabolism since these mice show a sex-dependent increase in sensitivity of Oxt-neuron-ablated male mice to weight gain after high-fat diet independent of increased food intake. Therefore, the increase in weight gain is a consequence of reduced metabolism and the disrupted energy balance in PVN [39,40]. Oxt-neuron-ablated mice have deficits in energy expenditure regulation to cold temperature but metabolic parameters or energy expenditure are unchanged at thermoneutrality [35,41,42].

## 4. Cold Stress Triggers the Oxytonic Contraction

Oxt regulates the physiological adaptation of the organism to challenging stimuli [6,30]. The lack of Oxtr leads to impaired thermogenesis with decreased core body temperature after acute exposure to cold [23]. Thermogenic challenge highlights c-Fos immunoreactivity of Oxt neurons in PVN [39]. Expression of c-Fos, or other immediate early gene products, by specific neurons is used as a marker of cell activation, making staining of these proteins a useful technique for functional anatomical mapping of neuroendocrine systems, such as Oxt, in response to specific stimuli [43]. Normal cold-induced thermogenesis was rescued in Oxtr−/− mice by reinserting Oxtr in hypothalamus with an adeno-associated virus-Oxtr vector [40].

Since Oxt is involved in thermoregulation, which is essential in energy balance and in the etiology of obesity [44], we measured the mRNA levels of Oxt and Oxtr in brain, bone, and brown adipose tissue of mice exposed to short time 6 h (6 h) and 5 days (5 d) cold stress (CS).

The evidence that Oxtr is upregulated consistently after short- and long-term CS in brain suggests that Oxt plays a main role in this tissue in response to CS challenge. Gene expression analysis shows that Oxt mRNA is upregulated in bone after 5 d CS, this indicate that Oxt is adaptive and important in restoring the homeostasis of the body in brain and bone. *Oxt* gene significantly decreased in BAT after 6 h CS but increased in bone after 5 d, supporting the concept that Oxt modulates energy and bone [6,24,45,46]. Oxt regulates the coordinated gene response to the paradigm of CS possibly acting as a master gene [6].

Interestingly, CS challenge for 6 h induced a significant increase of food intake in cold-stressed mice vs control mice but no changes in abdominal fat pad and body weight of the animals was observed. After a longer cold exposure for 5d, the abdominal fat pad weight was significantly decreased while food intake was enhanced by threefold in cold-stressed mice vs control mice. After an initial drop of body weight, it returned to control values. This means that cold-stressed mice are in good health and regained their body weight. The enhancement of food intake following CS is linked to Oxt signaling in brain as this factor is involved in the regulation of food consumption [6,10]. For example, Oxt levels in plasma decreased in obese mice after high-fat diet but increased in synaptotagmin-4-deficient mice, which protects against diet-induced obesity [47].

The evidence that Oxt KO mice develop obesity and impaired cold-induced thermogenesis without a change in food intake suggests that the lack of Oxt may reduce metabolic rate [23,24,39]. Nevertheless, administration of central Oxt reduces food intake in rats, an effect that is reversed by Oxt antagonist implying that Oxt may regulate appetite and energy intake [48,49,50].

Oxtr regulates the coordinated gene response to CS through a feed-forward loop in brain [5]. This is supported by the fact that in a regression study upon elimination of Oxtr expression gene data in brain, we report a loss of correlation in gene expression of mice at thermoneutrality versus gene expression of cold-stressed mice [6].

## 5. A New Hypothesis: Oxytocin Increases Skeletal Muscle Tonicity after Thermogenic Challenge

Oxt is a pleiotropic hormone with effects on the most diverse target organs. Indeed, Oxt is secreted by bone cells and bone cells express Oxtr [21], and Oxt is also required for muscle regeneration. Oxt levels decline with age but Oxtr expression remains unchanged in the old tissues [30]. Conversely, the exogenous administration of Oxt drives skeletal muscle regeneration and increases muscle tone [24,30], and Oxt does not cross the blood brain barrier [45]. Then in a second set of experiments, we showed that skeletal muscle properties are influenced by Oxt and the cross-talk between hormones of skeletal muscle with bone and brain allows adaptation of these tissues to CS as previously shown [10,51,52,53,54]. Skeletal muscle is also a source of heat production in cold-exposed humans and mice. This is achieved voluntarily as contractions from exercising muscles or involuntarily as contractions from shivering muscles. Indeed, one of the pathways activated by cold exposure is the involuntary activation of skeletal muscle movement. Shivering needs a significant amount of energy and, therefore, generation of ATP to sustain muscle contractions [55]. However, no data are available on the central and peripheral neuronal Oxt circuits involved in the involuntary muscle contractions following CS [56]. Adaptation to cold temperature and endurance physical exercise share similar metabolic characteristics increasing oxidative metabolism and lipolysis. Consequently, we hypothesized that CS could potentiate slow fiber type as seen after prolonged exercise and that this process is regulated by Oxt. The role of Oxt/Oxtr in the regulation of muscle phenotypes after exposure to cold is unknown. Moreover, in a stretch of mind and since Oxt contracts the myometrium [7,8], we generated the hypothesis that Oxt could virtually contract all the slow-twitch muscles having an enhancing muscle “tonic effect.” So, we investigated the effects of *Oxt* on soleus and TA muscles following CS. In rodent, four myosin heavy chain (MyHC) isoforms, such as MyHC1, 2A, 2X, and 2B, have been identified in skeletal muscle [53]. *MyHC1* is expressed in type 1, or slow-type, muscle fibers such as the soleus muscle. Types *MyHC2A, 2X*, and *2B* compose the fast-twitch of extensor digitorum longus or tibialis anterioris (TA) or Extensor digitorum longus [53,57]. Fibers expressing *MyHC2A* and *2X* show intermediate properties between type 1 and type *2B*. The *MyHC2X* fibers are fast-twitch glycolytic fibers, whereas type *2B* fibers show a more marked fast-twitch and glycolytic phenotype than type *2X* [58,59,60]. A severe cold stress-full condition of 4-week exposure at 4 °C increased the slow-type *MyHC1* content in the slow-type fiber of soleus muscle, while the intermediate-type *MyHC2A* content is reduced. [54]. In our experiments, we found a potentiation of the slow-twitch muscle phenotype after CS [54], as shown by the downregulation of fast-twitch glycolytic *myosin heavy-chain 2b (Mhc2b)* and the increase of the ratio of *myosin heavy chain 1 (slow-oxidative)/myosin heavy chain 2b* (fast-twitch glycolytic) expression in soleus but not in TA [10] consistent with the metabolic need of the slow-twitch oxidative muscle after CS. Long-term CS for 5 d also induced a downregulation of *Myhc2a* in TA muscle living unaltered the slow-twitch *Myhc1*. However, *Myhc2x* and *Myhc1* isoforms were also not significantly affected in either muscle phenotypes.

These findings suggest that 6 h and 5 d CS challenge specifically affects the slow-twitch muscle increasing the ratios *Myhc1/Myhc2b* and *Myhc1/Myhc2a* in soleus and TA muscles, respectively.

So, thermogenic stress shifts TA muscle toward the slow-twitch phenotype while potentiate the slow-twitch phenotype of soleus. In addition, *Oxtr* is highly expressed in slow-twitch muscle after 5 d CS but not in TA. We hypothesized a possible effect of Oxt signaling on the muscle phenotype following CS. Oxt up-regulation in skeletal muscle has been observed following androgen treatment [61], and Oxt regenerate muscle after sarcopenia [30,45]. Therefore, we hypothesized that Oxt exerts “tonic action” meaning an increase in the tone of the slow-twitch muscle similarly to what occurring in uterus [62]. Slow-twitch fibers are indeed more resistant than fast-twitching fibers to mechanical and metabolic insults so the oxytonic action on soleus muscle promoted by Oxt may lead to a more resistant phenotype against shivering thermogenesis. We also propose this Oxt circuit being mediated by the relation between brain and soleus through feed-forward/feed-back regulation [10]. Indeed, brain Oxt may up-regulate the response of the soleus at 6 h, while may down-regulate the brain–soleus intercommunication after 5 d CS: low circulating Oxt levels are required for a better response to 5 d CS challenge. Nevertheless, the Oxt signaling is maintained by the up-regulation of *Oxtr* gene found in soleus muscle at 5 d that balances the reduced level of circulating Oxt consistent with previous studies [30]. In conclusion, systemic Oxt can mediate tonic effects on slow-twitch muscle through upregulation of its receptor [10].

The genetic studies were confirmed by immunohistochemistry for Oxtr expression in hypothalamic PVN and in the hippocampus (HIPP) and by the measurement of plasmatic Oxt [11,63,64]. Histomorphometric analysis shown that thermogenic challenge increases the expression of Oxtr in PVN and HIPP. This trend is consistent with gene expression data in whole brain [6]. However, a different pattern of Oxtr expression is observed in SON, major site of Oxt secretion, where Oxtr is unchanged after 6 h CS, but decrease at 5 d. Plasmatic Oxt levels were unchanged after 6 h CS, while plasmatic Oxt levels were significantly reduced vs. control after 5 d CS. Oxt/Oxtr expression in PVN and plasma are consistent with in vitro studies [10]. Such modulation, together with Oxtr upregulation in soleus after CS, triggers the “oxytonic contraction”. Finally, Oxt upregulation in bone at 5 days CS may balance the decrease in plasmatic Oxt [6] (Figure 1).

The heart is a target organ for Oxt, and the heart expresses Oxtr. This means that Oxt acts on this organ in a paracrine/autocrine manner [65,66]. In our experiments of CS, we observed that Oxt may protect the cardiac muscle from necrotic process [6,11] possibly increasing its tonicity.

Emerging evidence also indicates that the *TRPV1* cation channels are a mediator of the pain signaling of Oxt [67,68]. The expression levels of *Oxtr* and *TRPV1* were both elevated following 5d CS in mice in soleus muscle but not in TA. These interactions can be responsible for the elevation of the intracellular calcium levels, in this muscle, and the cold-induced analgesia [11]. Overall, this study provides a new perspective over the physiological functions of Oxt that following CS may exert “oxytonic” action on slow-twitch muscle.

A recent study confirmed the involvement of Oxt in thermoregulation showing that Oxt expression in hypothalamus and Oxtr in adipose tissue were induced by cold exposure in mice [69]. Oxt stimulates brown adipocyte-specific gene expression after cold exposure, and exogenous Oxt increased body core temperature and decreased body weight gain in high-fat diet induced obese mice. Interestingly, Oxt induces browning of white adipose tissue stimulating thermogenesis in BAT and skeletal muscle. This means that Oxt is involved in both shivering and nonshivering thermogenesis, thus combating obesity and metabolic dysfunctions [69]. However, the expression of Oxt in different muscle phenotype and the Oxt-driven increases in muscle tone required for shivering thermogenesis need to be further investigated.

## 6. Oxytonic in Skeletal Muscle Physiology

Understanding of Oxtr regulation in metabolically active tissues and outside of pregnancy is of interest since the peptide displays potential antiobesity properties [66]. Recent studies have demonstrated a growth-regulating effect of Oxt mediated by Oxtr in different cell types like myoepithelial cells in mammary gland and uterine smooth muscle cells [53,70,71]. Oxtr is the unique receptor subtype expressed in a clonal culture of human myoblast and its activation by agonist binding stimulates the rate of myoblast fusion, and Oxt appears to be a paracrine/autocrine agent that might regulate the differentiation of human skeletal muscle cells. *Oxt* gene is also expressed by cultured myoblasts which suggests the presence of an intrinsic Oxt/Oxtr system [68]. Oxt appears to be a paracrine/autocrine agent that might regulate the differentiation of human skeletal muscle cells.

Even phenotypically diverse skeletal muscles show similar regulation of Oxtr in obese phenotype, suggesting a direct effect on these organs [72]. Interestingly, Oxtr is differentially expressed in various tissues according to the degree of obesity [66]. Elevated Oxtr protein determined by immunoblot was observed in epididymal adipose tissue of obese Zucker rats but was downregulated in subcutaneous fat with no change in retroperitoneal fat. In lean control, Oxt expression showed no changes between fat depots analyzed [72]. Differently in skeletal muscle, fiber type composition are determinants of Oxtr expression. In the skeletal muscle, the variation in Oxtr levels in quadriceps and soleus is related to fiber type composition and reflects different metabolic characteristics that exist among individual skeletal muscles [53,71]. An independent study showed that while quadriceps is highly oxidative in nature due to high mitochondrial content, soleus exhibits a more glycolytic phenotype [53]. This means that Oxt action in skeletal muscle is also associated with regulation of glucose other than lipid metabolism. Oxt is an age-specific circulating hormone, the absence of age-dependent changes of Oxtr in skeletal muscle is consistent with data obtained from gastrocnemius muscle [66]. In addition, cardiac muscle is an Oxt target organ [65,66]: Oxt is involved in the activation of cardioprotective mechanisms, inhibiting the development of hypertrophy, fibrosis, and inflammation in the myocardium, and Oxtr is expressed in cardiac muscle. A recent study showed that an animal model with leptin receptor defect, db/db mice, produces a state of obesity associated with reduced expression of cardiac Oxtr. Chronic treatment with Oxt prevents cardiomyopathy associated with obesity in these mice independently of hyperglycemia and hyperinsulinemia which suggests that Oxt has a direct effect on the cardiac muscle prioritized to obesity [29]. These data are consistent with our hypothesis of a direct effect of Oxt on skeletal muscle and heart that come prior to the effect on metabolism and food intake.

## 7. Oxytocin Is a Gastrointestinal Hormone: The Tonic Effect

Transcripts encoding Oxt and Oxtr have been reported in adult human gut, and Oxt affects intestinal motility. Oxt and Oxtr mRNA are present in adult rat and mouse gut and in precursors of enteric neurons from fetal rats. Enteric Oxt and Oxtr expression is developmentally regulated. Oxt acts on the bowel to modulate motility, secretion, and release of neurotransmitters and hormones. Oxt/Oxtr is present in most segments of the adult human gut, and Oxt immunoreactivity is present in myenteric and submucosal neurons of the human bowel [73,74,75,76,77]. The effects of Oxt on gastrointestinal motility are species dependent since Oxt accelerates [78,79] and Oxtr antagonists [77] delay gastric emptying in humans but Oxt delay gastric emptying in rats [80,81]. The effects of Oxt on human stomach is direct, and Oxt acts as a visceral analgesic and regulates the transmission of visceral nociceptive signals to the CNS, as we have shown in soleus muscle [11,82,83]. Since Oxt/Oxtr transcript and protein are both found in the enteric nervous system of mice, Oxt/Oxtr signaling do not depend on hypothalamic secretion of Oxt. Since many enteric neurons express Oxtr, Oxt in the systemic circulation exerts enteric effects. Therefore, hypothalamic and enteric sources of Oxt may act in synchrony [84].

Oxtr is expressed in the gastric smooth muscle and intestinal epithelium [20,22,23,24,25,26,27,28,29,30,31,32,33,34,35,36,37,38,39,40,41,42,43,44,45,46,47,48,49,50,51,52,53,54,55,56,57,58,59,60,61,62,63,64,65,66,67,68,69,70,71,72,73,74,75,76,77,78,79,80,81,82,83,84,85]. Recent studies show that cold water intake inhibits colonic motility in rats partially through Oxt/Oxtr signaling in the myenteric nervous system of rat pathway that is estrogen dependent [86,87]. Indeed, colon transit was decreased and Oxt inhibition of colonic contraction was enhanced in rats after cold water intake. Cold water intake also increased blood concentration of Oxt in an estrogen-dependent manner [87]. Oxt antagonists inhibited the effects of Oxt on colonic motility. Oxt was first described for its “tonic” regulation of smooth muscle in 2008, when studies on gastric motility showed that exogenous Oxt excited the circular muscle strips and isolated smooth muscle via Oxtr in gastric body of mice [20,74]. Oxt and Oxtr mRNA are expressed through the human GI tract, therefore, Oxt appears as a GI hormone that acts locally in the GI tract.

## 8. Oxytonic in Human Studies

Interestingly, studies on human subjects support the concept that low Oxt levels are associated with a loss of “body tonicity” [10,11,23,24]. For instance, in a cohort of overweight or obese African American veterans, urinary Oxt levels were associated with measures of glycemic status with higher value favoring a healthier metabolic phenotype. Moreover, several interesting associations in the cohort were observed with low Oxt being associated with increased impulsiveness and difficultness in social interactions. The levels of serum Oxt in obese subjects were also lower than those in normal weight subjects. Serum Oxt levels are negatively correlated with different metabolic parameters such as total cholesterol and triglycerides among others [47,88]. The mechanisms underlying decreased Oxt levels in obesity and T2DM remain unknown. This study on human subjects is limited by its cross-sectional design and does not explain the physiological reason of low plasma Oxt in the metabolic syndrome [89]. Nevertheless, a similar study in obese and diabetic children confirms the concept that low Oxt characterizes this metabolic profile. Serum Oxt level was significantly decreased in obese children compared with controls and also lower in obese children with metabolic syndrome compared with those without. These studies in children are consistent with those on adult subjects. Oxt level in obese children is inversely proportional to the severity of metabolic phenotype. Decreased Oxt levels lead to impaired thermoregulation and increased food consumption in obese children [90].

Both adipose and muscle tissue contribute to the establishment of metabolic phenotype. Young female athletes are considered in a higher energy expenditure state than nonathletes. Oxt is a signal for energy availability and is associated with other measures of energy homeostasis in young athletes. Oxt secretion is positively correlated with measures of energy availability as weight and body mass index and energy expenditure in amenorrheic athletes, that is considered the highest energetic state. These subjects have a higher lean mass and skeletal muscle functionality than eumenorrheic athletes or nonathlete consistent with our hypothesis that Oxt augments the muscular functions. Interestingly, no significant relationship was shown between Oxt levels and energy measures in nonathletes young women [50,91].

However, postprandial peripheral Oxt levels in females are lower compared to fasting levels of Oxt independently of menstrual cycle or meal size because endogenous Oxt levels are a satiety signal and Oxt plays a role in the regulation of food intake [92].

Amenorrheic athletes in whom nocturnal levels of Oxt are lower than in eumenorrheic athletes or nonathletes showed impaired bone microarchitecture at the tibia and radius, particularly at the nonweight-bearing radius, although these evidences can be biased by the low estrogen levels of amenorrheic athletes since estradiol stimulates Oxt secretion [93]. Oxt KO mice show high bone mass consequent to the obesity and high leptin, while only intraperitoneal administration of Oxt may be anabolic to bone for its tissue-specific effects [21,45,94]. Recent studies show that Oxt increases acutely in male and female athletes after intense/endurance training, showing, e.g., an evident increase in Oxt plasmatic levels after marathon [95,96,97].

## 9. Oxytonic in Human Diseases: Prader–Willy Syndrome and Autistic Spectrum Disorder

Prader–Willy syndrome (PWS) is a rare genetic disorder characterized by a behavioral phenotype including hyperphagia, social deficits, and obsessive-compulsive tendencies. Oxt may play a role in PWS; indeed, postmortem assays showed that PWS males showed reduced *Oxtr* gene expression and density in PVN [98]. To evaluate Oxt biology in PWS, overnight fasting plasma Oxt levels in children with PWS having genetic confirmation with healthy unrelated siblings without PWS was examined [99]. Plasmatic Oxt are higher in PWS patients compared with unaffected siblings and the diagnosis of PWS predicted Oxt levels [100]. The symptoms of hyperphagia and behavior seen in PWS may be related to the disruption of Oxt responsivity or feedback in PVN. This phenotype is the striking mirror image of the phenotype of cold-stressed mice in our model of thermogenic stress [11] where Oxtr mRNA increased in PVN, whereas Oxt decreased in plasma after CS in mice. Oxtr mRNA was also higher in slow-twitch muscle of cold-stressed mice. This led to increased tonicity of the slow-twitch muscles [11]. Since an important hallmark of PWS is, in fact, hypotonicity at birth [101,102], we hypothesized that this hypotonic state led to late onset obesity at older age, although this hypothesis needs to be integrated by an extensive study of Oxt/Oxtr in skeletal muscles of PWS patients that is now lacking. PWS is a rare neurodevelopmental disorder due to absent expression of paternally active genes on chromosome 15q11.2-q13 region. PWS patients are hypotonic at birth and have feeding problems as infants, global developmental delay, hyperphagia and obesity at childhood, behavioral problems, hypogonadism, and short stature that suggest abnormalities of the hypothalamic–pituitary axis. When PWS patients are treated with growth hormone, improvement is seen with increased muscle size and decreased fat mass [103,104,105]. Understanding the significance of Oxt in PWS patients may be an important step in developing new treatments.

Although a recent study reported higher plasma Oxt in adolescent girls with PWS compared to male control, no differences are reported in adult patients with PWS [106]. This is similar between male and female control differences. The posterior pituitary releases Oxt in response to stimulation like birthing, eating, and a variety of challenging situations. Children with PWS were found to have significantly increased morning fasting plasma levels of Oxt as compared to unrelated siblings. Another study reported elevated Oxt levels in cerebrospinal fluid (CSF) of PWS. Plasma Oxt is aligned with CSF Oxt levels, suggesting that plasma Oxt levels act as a surrogate for central nervous measure of Oxt activity [105,107]. This relation was independent of body fat content or BMI and this increase may be caused by a change in peripheral Oxt synthesis at the tissue levels due to increased muscle or bone mass [100]. Oxtr mRNA is also decreased in temporal cortex of autistic spectrum disorder (ASD) that is also component of PWS. A reduced expression of Oxtr in PVN, in the dysfunctional Oxt system of PWS patients, lead to increased Oxt secretion by the posterior pituitary due to the loss of negative feedback [99]. Conversely, in our model of cold stress, where the Oxt system is functioning appropriately, the extreme low temperature increased Oxtr in PVN and in soleus muscle causing the decrease of Oxt in plasma. This mechanism triggers the “oxytonic contractions” that potentiate the slow-twitch muscle for physical reaction to challenging situations [10,11]. The prolonged exposure to high levels of endogenous Oxt may downregulate Oxtr reducing the effectiveness of Oxt, in this case, Oxt may stimulate the receptor for the related molecule vasopressin. Under these conditions, exogenous Oxt might have the potential to trigger behaviors like increased agitation and defensive aggression, which are associated with the stimulation of vasopressin receptor [108].

Two mouse models deficient in genes involved in PWS, necdin and Magel2, show similar Oxt deficiency to that seen in PWS patients [109,110]. Postnatal central administration of Oxt has been shown to rescue the feeding deficiency and social deficit in Magel2 KO mice. This finding shows that the administration of Oxt may have some effects on the abnormality in the Oxt system in PWS [108]. Nevertheless, treatment with exogenous Oxt during clinical trials in PWS patients lacked of positive effects, since pharmacological therapy with Oxt does not reverse the nature of Oxt abnormality. Indeed, Oxt is produced in an inactive form of 12 amino acids and is then cleaved and released as a 9 amino acids active form. It is the active form that binds to the receptors. Individuals with PWS have increased levels of Oxt in plasma. This is inconsistent with previous data reporting a negative association between plasmatic levels of Oxt and the severity of metabolic syndrome [99,111]. This data can reflect high levels of inactive form and not enough of the active form, alternatively, Oxtr could be suppressed in PWS after a genetic or methylation defect. In support of this hypothesis, PWS patients show a decrease in Oxtr gene in PVN [112].

In a different study, intranasal Oxt was administered to 24 children for 5 days followed by 4 weeks washout and then again for 5 days [113]. Improvements in behavior and decreased food intake was registered from day 3 to day 6 of treatment while the drug effect diminished at day 14 meaning that low dose Oxt is safe for individuals with PWS. In mice, long-term administration of Oxt results in reduced Oxtr in brain [114] that can be the case for PWS patients in this study. Exposure to Oxt leads to desensitization of Oxtr to protect cells from overstimulation [99]. This is reported in the majority of G protein-coupled receptors including Oxtr [115]. However, these receptors are recycled back to the cell surface once Oxt levels have decreased [115]. Therefore, a longer therapeutic treatment with Oxt could cause the Oxtr to be internalized into the cells and without being recycled to the cell surface. Therefore, longer term and dose-response studies are necessary in this population [113,116,117,118]. Further studies will be necessary to measure Oxt/Oxtr at the tissue levels in skeletal muscle of PWS individuals to understand if dysfunctional “oxytonic contractions” could be part of PWS syndrome.

## 10. A Genetic Approach: The MAGE Family of Genes

Autism spectrum disorder (ASD), is a developmental brain disorder that include autism per se, Asperger syndrome as well as other severe developmental disorders not otherwise specified [119]. NECDIN and MAGEL2 are members of the MAGE gene family and belong to paternally expressed genes on chromosome 15 that are not expressed in patients with PWS [120,121]. Knockout experiments in mice suggests that deficiency in NECDIN and MAGEL2 contributes to PWS including a reduced number of hypothalamic neurons producing Oxt and gonadotropin-releasing hormone [15].

The MAGE genes encode multifunctional proteins involved in cell cycle, differentiation, and survival [122]. Some of these genes, called type I, are expressed exclusively in germ cells, whereas others, type II, are expressed predominantly in the nervous system [123]. MAGED1 inactivation in mice reduced neuronal apoptosis, delay muscle regeneration, and disrupt the circadian clock function [15,124]. Loss of MAGED1 leads to reduced social interactions, a behavioral phenotype that resemble autism, late onset obesity, and hyperphagia. MAGED1 is expressed at high levels in PVN and SON, which are the major site of Oxt production. The brain of MAGED1 mice contained a reduced Oxt-neurons population and less mature Oxt. Neurons containing the precursors and intermediate forms of Oxt were expressed at normal levels suggesting a lack of maturation of Oxt intermediate forms. The production of mature AVP appears to be normal. The level of mature Oxt is also reduced after loss of MAGEL2 [110], indicating that MAGED1 and MAGEL2 are both essential for the production of normal levels of OXT in hypothalamus. Loss of MAGED1 resulted also in mild obesity, hyperphagia, and hypoactivity, although no study so far investigated muscle tone. The administration of Oxt rescued the deficit in social memory of these mice.

So, MAGED1 is required for Oxt processing or stability, and in human, MAGED1 is involved in autism etiology or cause a neurodevelopmental condition similar to PWS [15].

The genes in the PWS critical region include MAGEL2 that encode for a protein important for endosomal protein trafficking [125]. Individuals with truncation region of MAGEL2 manifest a phenotype that has remarkable overlaps with PWS. However, these patients are phenotypically distinct from patients with classic PWS, and so, the syndrome caused by MAGEL2 point mutation was renamed from PWS to Schaaf-Yang syndrome. Individuals with Schaaf-Yang syndrome do not develop hyperphagia and obesity but manifest contracture of interphalangeal joints and ASD. MAGEL2 KO mice showed several behavioral abnormalities, plus fat infiltration in muscle, abnormal neurotransmitter signaling, and changes in volume of certain brain areas [126]. This adds further credence to the involvement of Oxt in muscle physiology and solicits interest to study if Oxt treatment would rescue also this phenotype [19].

## 11. Conclusions

Going back to the principle of maximum parsimony, the findings described in this review show how Oxt exerts two or more additional functions other than uterotonic as regulating energy metabolism and potentiating muscular function. The posterior pituitary releases Oxt in response to a variety of challenging situations and evolutionarily changes. These physiological effects of Oxt are consistent with augmented energetic need at time of labor when Oxt secretion is higher and augmented muscular strength for the protection of off-spring at the face of a physical threat. The data described needs to be further confirmed by extensive studies on human subjects in physiological and pathological conditions.

## Figures and Tables

**Figure 1 ijms-21-05144-f001:**
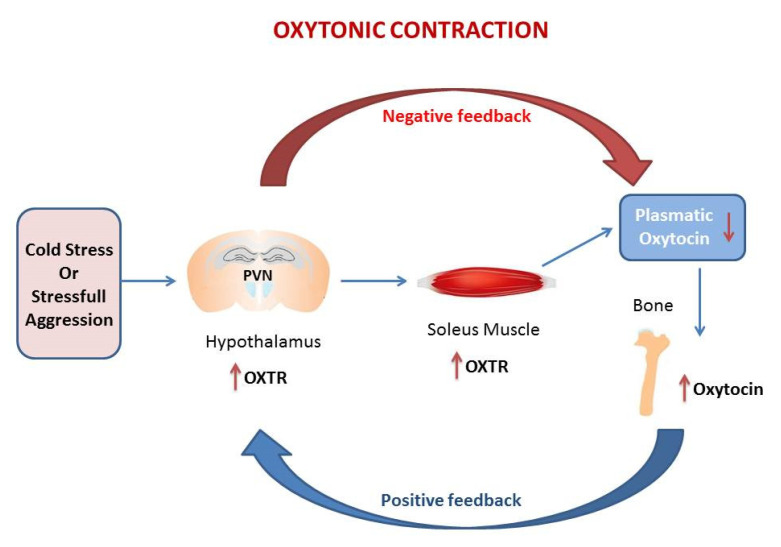
The figure in the manuscript was drawn by the author. The oxytonic contraction: in our model of cold stress, the extreme low temperature increases Oxytocin receptor (Oxtr) in PVN and at the tissue levels in the soleus muscle but drive the decrease of Oxytocin (Oxt) in plasma. The increase of Oxt in bone balance the decrease of plasmatic Oxt. This mechanism triggers the oxytonic contractions that potentiate the slow-twitch muscle to ensure the proper physical reaction to challenging situations. The red arrow represents the negative feedback that decreases Oxt in plasma following the increases of Oxtr in PVN. The blu arrow represents the positive feedback that increases Oxtr in PVN following the drop of Oxt in plasma.

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
