# Peer review of "The New Frontier in Oxytocin Physiology: The Oxytonic Contraction"

_ijms, 2020, doi:10.3390/ijms21145144_

Round 1

Reviewer 1 Report

This an extensive and heavily referenced article on OXT based on the authors' many years of research on the subject. The data is comprehensive. The major problem is presentation. This can be broken down into 2 areas:

  1. English - as one who has written in a non-native language, this reviewer is aware of the difficulty of doing such. The English here is reasonable but there are areas that are hard to follow. This is not a criticism, but a fact. An English editor is needed.
  2. There is no overall plan for the reader to follow in easily understood language to know what the authors are trying to say. They begin by inferring that OXT has pleiotropic effects. They should then say "we will present data that OXT has several effects: metabolism, muscle, hunger, etc." In such a away the reader is guided as to what to expect. As it stands there is no sense of overall organization to the paper.

Specific comments by line number for the first half of the paper:

  1. Lines 36-38 - confused as written. What you want to say is that genes have many effects, hence the need for less genes, correct?
  2. Line 38 - start a new paragraph. Otherwise, there is no connection with what precedes it.
  3. Lines 53-55  - there is no logical connection between what you say at the onset and with what you conclude. Please fix.
  4. Lines 57ff - here you discuss OXT and lower appetite but this not at all connected to what precedes it. This section is out of place as it stands. Perhaps write here that you will now discuss the myriad effects of OXT beginning with appetite and metabolism.
  5. Line 76 - "depotentiation" - what does that mean? loss of function?
  6. Lines 57-86 - you are mixing many subjects together in one paragraph. This paragraph is hard to understand.
  7. Line 90 - what is "cre" expression?
  8. Line 99 - "undercover" - what is that?
  9. Lines 105-106, 114-115 - I do not understand what is written
  10. Line 119 - c-Fos  - you mention this but do not tell us what fos does in the context of OXT.
  11. Line 137 - recapitulated - do you mean regained their weight?
  12. Lines 143-145 - bone, brain, muscle are all mixed up together. Hard to follow
  13. Line 168ff - what does the mych gene do? There is no explanation.
  14. Lines 198-200 - suddenly out of nowhere the heart is mentioned without any context. This has nothing to do with anything.
  15. Lines 240-245 - hard to follow and out of context

Author Response

  1. English - as one who has written in a non-native language, this reviewer is aware of the difficulty of doing such. The English here is reasonable but there are areas that are hard to follow. This is not a criticism, but a fact. An English editor is needed.
  2. There is no overall plan for the reader to follow in easily understood language to know what the authors are trying to say. They begin by inferring that OXT has pleiotropic effects. They should then say "we will present data that OXT has several effects: metabolism, muscle, hunger, etc." In such a away the reader is guided as to what to expect. As it stands there is no sense of overall organization to the paper.

The reviewer is right and a more clear explanation fo what is the scope of this review has been written. In lines 39-45 I explain that in this review we described  the several effects of Oxt in different organs with a special attention to skeletal muscle and our recent data of cold stress. Indeed in this review I also introduced our most recent hypothesis that Oxt may increase skeletal muscle tone. We recently shown that Oxtr increases in Soleus muscle after cold stress challenge in mice (Camerino , Frontiers 2019). Oxt and Oxtr deficient mice present metabolic phenotype but are not hyperphagic. This let us hypothesized that the increased body weight in this murine model was caused by loss of skeletal muscle tone rather that decreased food consumption. This is a paradigm shifting concept and I hope now is more clear to the reader.

This concept has been better explained also in line 53-62.

Specific comments by line number for the first half of the paper:

  1. Lines 36-38 - confused as written. What you want to say is that genes have many effects, hence the need for less genes, correct?

The reviewer is right and the sentence has been rearranged and a new reference has been added.

  1. Line 38 - start a new paragraph. Otherwise, there is no connection with what precedes it

The reviewer is right and the parapragh has been rearranged.

  1. Lines 53-55  - there is no logical connection between what you say at the onset and with what you conclude. Please fix.

The reviewer is right and the all paragraph has been rearranged. New sentences has been added in lines 66-68-69.

  1. Lines 57ff - here you discuss OXT and lower appetite but this not at all connected to what precedes it. This section is out of place as it stands. Perhaps write here that you will now discuss the myriad effects of OXT beginning with appetite and metabolism.

The reviewer is right and a new sentence has been added in line 65-66.

  1. Line 76 - "depotentiation" - what does that mean? loss of function?

The reviewer  is right and “depotentiation”, has been corrected to “Loss of function”.

  1. Lines 57-86 - you are mixing many subjects together in one paragraph. This paragraph is hard to understand.

The reviewer is right and the paragraph has been rearranged. A new explanation has been added in lines 87-88 and 91-92.

  1. Line 90 - what is "cre" expression?

The reviewer  is right and a short explanation of Cre-Lox has been added to line 113 together with a new reference.

  1. Line 99 - "undercover" - what is that?

In line 99 the word “undercover” has been corrected to “These findings show”.

  1. Lines 105-106, 114-115 - I do not understand what is written

The reviewer is right and the line 121-124 has been rearranged. The lines  114-115 has been deleted because redundant.

  1. Line 119 - c-Fos  - you mention this but do not tell us what fos does in the context of OXT.

The reviewer is right and a new sentence has been added to lines 142-148 together  with a new reference.

  1. Line 137 - recapitulated - do you mean regained their weight?

In Line 137 the word “recapitulated” has been changed to “regained”.

  1. Lines 143-145 - bone, brain, muscle are all mixed up together. Hard to follow

The reviewer is right and a short introduction has been added to this paragraph at line 166.

  1. Line 168ff - what does the mychgene do? There is no explanation.

The reviewer is right and explanation for  mych gene has been added to line 197-205.

  1. Lines 198-200 - suddenly out of nowhere the heart is mentioned without any context. This has nothing to do with anything.

The reviewer is right and the sentence has been rearranged. A new introductory  sentence  has been added at line 213.

  1. Lines 240-245 - hard to follow and out of context

The reviewer is right and the paragraph has been rearranged. New sentences have been added to line 255 and 262-263.

However I would like to underline that  in this paragraph I’m trying to explain the concept that Oxt affects directly the skeletal muscle. This is a paradigm shifting concept, so I hope it is enough clear and can be accepted by these reviewers.

Reviewer 2 Report

In the present paper dr Camerino reviews new findings about the role of oxytocin in the regulation of body weight and energy balance. The main aspects covered in the paper include the phenotype of oxytocin knockout mice, changes in oxytocin system in the experimental models of obesity, characteristics of mice lacking oxytocin specifically in hypothalamic neurons, effect of cold stress on oxytocin expression, effect of oxytocin and cold stress on the potentiation of slow-twitch muscle fiber type, role of oxytocin expressed in skeletal muscles and gastrointestinal tract, alterations of oxytocin system in human obesity, role of oxytocin in Prader-Willi syndrome and autism spectrum disorders.

The topic is very interesting and a lot of important findings are presented and analyzed. However, there are also many concerns to be addressed. In particular, the manuscript requires thorough lamguage revision.

  • Line 16: “Oxt potentiate…” should be corrected to “Oxytocin potentiates…”
  • Line 43: “…and release it in the bloodstream…” should be corrected to: “…and release to the bloodstream…”.
  • Line 52: oxytocin abbreviation should be unified throughout the text (Oxt or OXT).
  • Line 59: “in contrast with the expectations…” should be corrected to: “in contrast to the expectations…”.
  • Line 60: “Oxt decreases food intake increasing leptin concentration in plasma” should be corrected to: “Oxt decreases food intake by increasing leptin concentration in plasma”.
  • Line 73: “the role of Oxt on metabolism and energy” should be corrected to: “the effect of Oxt on metabolism and energy”.
  • Line 77, the phrase “intra-muscular adipose tissue” needs clarification. Do the authors mean exactly adipose tissue (adipocytes) or lipid deposition in skeletal muscle cells?.
  • Lines 81/82: db/db mice lack the long isoform of the leptin receptor (Ob-Rb) and as such are leptin-resistant. Please explain.
  • Lines 95/96, the sentence is confusing. “Lower body temperature” and “higher heat at the skin” are contradictory. What is “higher heat at the skin”?
  • Lines 145/146, please explain the sentence “Oxt augments the physiological function of the body”.
  • Line 174: “let as hypothesized” should be corrected to: ““let as to hypothesize” or simply “we hypothesized”.
  • Line 177, the phrase: “tonic action” is ambiguous. It could be understood as the increase in muscle tone but also as any case of constant background activity of the hormone/mediator.
  • Line 182: “response to soleus” should be corrected to: “response of the soleus”.
  • Line 217: “grown-regulating” should be corrected to: “growth-regulating”.
  • Line 220: “Oxt appears to be as a paracrine/autocrine agent” should be corrected to: “Oxt appears to be a paracrine/autocrine agent”.
  • Lines 229-230, the sentence: “In lean control Oxt expression show no changes in any fat district analyzed”. It is unclear what change is mentioned here because no interventions are described. Maybe, the author mean no difference between fat depots rather than no changes over time. Anyway, the text should be clarified.
  • Line 231: “are determinant” should be „are determinants” or “is the determinant”.
  • Line 232: “fibers type composition” should be: “fiber type composition”.
  • Line 234: “quadriceps are highly oxidative” should be corrected to: “quadriceps is highly oxidative”.
  • Line 238: “Oxt plasmatic levels decreases” should be corrected to: “Oxt plasmatic level decreases”.
  • Title of section 6: “human trials” should be changed to: “human studies” because mostly observational rather than interventional studies are described.
  • Line 281: “metabolic parameters as total cholesterol” should be corrected to: “metabolic parameters such as total cholesterol”.
  • Lines 281/282: “The mechanisms…remains unknown” should be either “The mechanisms…remain unknown” or “The mechanism…remains unknown”.
  • Lines 282-284: data about Oxt levels in animal models of obesity should not be presented in the section focused on human studies.
  • Lines 285/286: “…low Plasmatic Oxt and metabolic syndrome” should be: “low plasma Oxt in the metabolic syndrome”.
  • Line 290: “adults subjects” should be corrected to: “adult subjects”.
  • Lines 293-297: this section focused on animal experiments should be shifted to the other part of the paper.
  • Line 296: “an effects that is reversed” should be corrected to: “an effect that is reversed”.
  • Lines 304-205: “non athletes young women” should be: “non-athlete young women”.
  • Line 306: do the author means that postprandial Oxt level is lower than fasting Oxt level?
  • Line 307: “cicle” should be corrected to: “cycle”.
  • Line 308: “in which” should be corrected to: “in whom” regarding human subjects. Furthermore, in the same line “than eumenorrheic” should be: „than in eumenorheic”.
  • Line 319: “essays” should be “assays”.
  • Lines 322/323, the sentence: “Plasmatic Oxt are higher in PWS patients compared with unaffected siblings but the diagnosis of PWS predicted Oxt levels” is unclear. The second part of this sentence is consistent with the first one, so the word “but” suggesting controversies is not suitable.
  • Line 332: “dues to” should be „due to”.
  • Lines 348/349, the sentence: “PWS male patients show also decreased Oxtr expression in lymphoblasts [108]” – decreased vs. females or decreased vs. healthy non-PWS males?
  • Line 350: “A reduced number of Oxtr in PVN”; “number” should be corrected to: “amount” or “expression”.
  • Lines 350-353: it is suggested that increase in Oxt synthesis/secretion may be the consequence of reduced Oxtr expression. However, the opposite possibility that Oxtr deficiency results in the compensatory up-regulation of Oxt should also be considered. Besides, regarding Oxt and Oxtr expression in the hypothalamus authors often refer to negative feedback or loss of negative feedback. Is there any evidence that Oxt inhibits its own secretion in the negative feedback manner?
  • Line 367: “It is this active forms” should be corrected to: “It is the active form”.
  • Line 388: “other severe developmental disorder” should be “disorders”
  • Line 396: “expressed in predominantly” should be: “expressed predominantly”.
  • Lines 396/397: did MAGED1 deficiency reduce or stimulate neuronal apoptosis?
  • The recent study on the topic (Mol Cell Endocrinol. 2020; 514:110903. doi: 10.1016/j.mce.2020.110903) should be cited and discussed.

Author Response

  • Line 16: “Oxt potentiate…” should be corrected to “Oxytocin potentiates…”

Revised as requested

  • Line 43: “…and release it in the bloodstream…” should be corrected to: “…and release to the bloodstream…”.
  • Revised as requested
  • Line 52: oxytocin abbreviation should be unified throughout the text (Oxt or OXT).
  • The Oxytocin abbreviation has been unified to Oxt.
  • Line 59: “in contrast with the expectations…” should be corrected to: “in contrast to the expectations…”.
  • Revised as requested.
  • Line 60: “Oxt decreases food intake increasing leptin concentration in plasma” should be corrected to: “Oxt decreases food intake by increasing leptin concentration in plasma”.
  • Revised as requested.
  • Line 73: “the role of Oxt on metabolism and energy” should be corrected to: “the effect of Oxt on metabolism and energy”.
  • Revised as requested.
  • Line 77, the phrase “intra-muscular adipose tissue” needs clarification. Do the authors mean exactly adipose tissue (adipocytes) or lipid deposition in skeletal muscle cells?.

The reviewer is right. In these review we presented this as preliminary data and the study of Oxt in skeletal muscle and adipocyte accumulation is incomplete and need further investigation. This concept has been explained at line 95-97.

Lines 81/82: db/db mice lack the long isoform of the leptin receptor (Ob-Rb) and as such are leptin-resistant. Please explain.

  • A new explanation has been added to line 81.
  • Lines 95/96, the sentence is confusing. “Lower body temperature” and “higher heat at the skin” are contradictory. What is “higher heat at the skin”?

The reviewer is right and the correct reference has been added to this sentence.

The core body temperature in these mice diverge from temperature measured at the surface or skin. This explanation has been reported at line 114.

  • Lines 145/146, please explain the sentence “Oxt augments the physiological function of the body”.

The reviewer is right. The sentence “Oxt augments the physiological function of the body”, has been corrected to “Oxt drives  skeletal muscle regeneration and increases muscle tone”.

  • Line 174: “let as hypothesized” should be corrected to: ““let as to hypothesize” or simply “we hypothesized”.
  • Line 174 has been revised to “We Hypothesized”.
  • Line 177, the phrase: “tonic action” is ambiguous. It could be understood as the increase in muscle tone but also as any case of constant background activity of the hormone/mediator.

The rewiever is right and Tonic action has been explain to increase in muscle tone. This is a completely new concept and I hope that now is clear.

  • Line 182: “response to soleus” should be corrected to: “response of the soleus”.
  • Revised as requested.
  • Line 217: “grown-regulating” should be corrected to: “growth-regulating”.
  • Revised as requested.
  • Line 220: “Oxt appears to be as a paracrine/autocrine agent” should be corrected to: “Oxt appears to be a paracrine/autocrine agent”.
  • Revised as requested.
  • Lines 229-230, the sentence: “In lean control Oxt expression show no changes in any fat district analyzed”. It is unclear what change is mentioned here because no interventions are described. Maybe, the author mean no difference between fat depots rather than no changes over time. Anyway, the text should be clarified.

The reviewer is right and this  sentence has been corrected to “in lean control …Between fat depot analyzed”. The correct reference has been added.

  • Line 231: “are determinant” should be „are determinants” or “is the determinant”.
  • Line 231 has been revised to “Are determinants”.
  • Line 232: “fibers type composition” should be: “fiber type composition”.
  • Revised as requested.
  • Line 234: “quadriceps are highly oxidative” should be corrected to: “quadriceps is highly oxidative”.
  • Revised as requested.
  • Line 238: “Oxt plasmatic levels decreases” should be corrected to: “Oxt plasmatic level decreases”.
  • Revised as requested.
  • Title of section 6: “human trials” should be changed to: “human studies” because mostly observational rather than interventional studies are described.
  • Revised as requested.
  • Line 281: “metabolic parameters as total cholesterol” should be corrected to: “metabolic parameters such as total cholesterol”.
  • Revised as requested.
  • Lines 281/282: “The mechanisms…remains unknown” should be either “The mechanisms…remain unknown” or “The mechanism…remains unknown”.
  • Line 281 has been revised as “The mechanisms remain”.
  • Lines 282-284: data about Oxt levels in animal models of obesity should not be presented in the section focused on human studies.

The reviewer is right and lines 281-282 have been added to chapter 3 lines 155-156.

  • Lines 285/286: “…low Plasmatic Oxt and metabolic syndrome” should be: “low plasma Oxt in the metabolic syndrome”.
  • Revised as requested.
  • Line 290: “adults subjects” should be corrected to: “adult subjects”.
  • Revised as requested.
  • Lines 293-297: this section focused on animal experiments should be shifted to the other part of the paper.

The reviewer is right and lines 293-297 have been moved to chapter 3 lines 157-160.

  • Line 296: “an effects that is reversed” should be corrected to: “an effect that is reversed”.
  • Revised as requested.
  • Lines 304-205: “non athletes young women” should be: “non-athlete young women”.
  • Revised as requested.
  • Line 306: do the author means that postprandial Oxt level is lower than fasting Oxt level?

The reviewer is right and post prandial Oxt levels are lower than fasting Oxt because Oxt is s satiety hormone. This explanation has been added to line 324-326.

  • Line 307: “cicle” should be corrected to: “cycle”.
  • Revised as requested.
  • Line 308: “in which” should be corrected to: “in whom” regarding human subjects. Furthermore, in the same line “than eumenorrheic” should be: „than in eumenorheic”.
  • Revised as requested.
  • Line 319: “essays” should be “assays”.
  • Revised as requested.
  • Lines 322/323, the sentence: “Plasmatic Oxt are higher in PWS patients compared with unaffected siblings but the diagnosis of PWS predicted Oxt levels” is unclear. The second part of this sentence is consistent with the first one, so the word “but” suggesting controversies is not suitable.
  • In lines 322/323 the word “but” has been changed to “and”.
  • Line 332: “dues to” should be „due to”.
  • Revised as requested.
  • Lines 348/349, the sentence: “PWS male patients show also decreased Oxtr expression in lymphoblasts [108]” – decreased vs. females or decreased vs. healthy non-PWS males?

The reviewer is right. This sentence has been deleted because redundant with the rest of the paragraph.

  • Line 350: “A reduced number of Oxtr in PVN”; “number” should be corrected to: “amount” or “expression”.
  • In line 350 the word “number” has been corrected to “expression”.
  • Lines 350-353: it is suggested that increase in Oxt synthesis/secretion may be the consequence of reduced Oxtr expression. However, the opposite possibility that Oxtr deficiency results in the compensatory up-regulation of Oxt should also be considered. Besides, regarding Oxt and Oxtr expression in the hypothalamus authors often refer to negative feedback or loss of negative feedback. Is there any evidence that Oxt inhibits its own secretion in the negative feedback manner?

The reviewer is right. For example Oxt can decrease with aging while Oxtr stays in the old cells this is why is more plausible that Oxtr upregulation drive the decrease of circulating Oxt as we shown in our results and in lines 371-372.

Moreover a  better explanation has been added to lines 371-373.

However a detailed explanation of this concept is also explained at lines 398-401.

  • Line 367: “It is this active forms” should be corrected to: “It is the active form”.
  • Revised as requested.
  • Line 388: “other severe developmental disorder” should be “disorders”
  • Revised as requested.
  • Line 396: “expressed in predominantly” should be: “expressed predominantly”.
  • Revised as requested.
  • Lines 396/397: did MAGED1 deficiency reduce or stimulate neuronal apoptosis?

The reviewer is right: MAGED1 inactivation reduces neuronal apoptosis. The right reference has been added to line 417.

  • The recent study on the topic (Mol Cell Endocrinol. 2020; 514:110903. doi: 10.1016/j.mce.2020.110903) should be cited and discussed.

I thank you very much this reviewer for mentioning such interesting article.

This article has been discussed at line 251-260 and the new reference has been added.

Round 2

Reviewer 1 Report

Once again there are parts of the paper early on that are not well organized. using multiple headings may help

Author Response

The reviewer is right. New  headlines have been added to lines 46 and 146. We hope the chapters are easier to read and shorter in length.

Reviewer 2 Report

The manuscript has been revised according to the reviewers' comments.

Author Response

Thank you very much